# Noninvasive ventilation in critically ill very old patients with pneumonia: A multicenter retrospective cohort study

Bruno A. M. P. Besen[1,2]*, Marcelo Park[2], Otávio T. Ranzani[3,4]

**1** Intensive Care Unit, Hospital da Luz, São Paulo (SP), AMIL, United Health Group (UHG), São Paulo, Brasil, **2** Medical Intensive Care Unit, Disciplina de Emergências Clínicas, Departamento de Clínica Médica, Hospital das Clínicas HCFMUSP, Faculdade de Medicina, Universidade de São Paulo, São Paulo, SP, Brasil, **3** Barcelona Institute for Global Health, ISGlobal, Barcelona, Spain, **4** Pulmonary Division, Heart Institute (InCor), Hospital das Clinicas HCFMUSP, Faculdade de Medicina, Universidade de Sao Paulo, São Paulo, Brazil

* brunobesen@yahoo.com.br

## Abstract

### Background

Very old patients ($\geq$ 80 years-old, VOP) are increasingly admitted to intensive care units (ICUs). Community-acquired pneumonia (CAP) is a common reason for admission and the best strategy of support for respiratory failure in this scenario is not fully known. We evaluated whether noninvasive ventilation (NIV) would be beneficial compared to invasive mechanical ventilation (IMV) regarding hospital mortality.

### Methods

Multicenter cohort study of VOPs admitted with CAP in need of IMV or NIV to 11 Brazilian ICUs from 2009 through 2012. We used logistic regression models to evaluate the association between the initial ventilatory strategy (NIV vs. IMV) and hospital mortality adjusting for confounding factors. We evaluated effect modification with interaction terms in pre-specified sub-groups.

### Results

Of 369 VOPs admitted for CAP with respiratory failure, 232 (63%) received NIV and 137 (37%) received IMV as initial ventilatory strategy. IMV patients were sicker at baseline (median SOFA 8 vs. 4). Hospital mortality was 114/232 (49%) for NIV and 90/137 (66%) for IMV. For the comparison NIV vs. IMV (reference), the crude odds ratio (OR) was 0.50 (95% CI, 0.33–0.78, p = 0.002). This association was largely confounded by antecedent characteristics and non-respiratory SOFA ($_{adj}$OR = 0.70, 95% CI, 0.41–1.20, p = 0.196). The fully adjusted model, additionally including $P_{a}o_{2}/F_{i}o_{2}$ ratio, pH and $P_{a}co_{2}$, yielded an $_{adj}$OR of 0.81 (95% CI, 0.46–1.41, p = 0.452). There was no strong evidence of effect modification among relevant subgroups, such as $P_{a}o_{2}/F_{i}o_{2}$ ratio $\leq$ 150 (p = 0.30), acute respiratory acidosis (p = 0.42) and non-respiratory SOFA $\geq$ 4 (p = 0.53).

**Data Availability Statement:** Due to privacy and data protection regulations in Brazil and considering that data refers to personal information of the participants which cannot be ensured to be

anonymized, data cannot be made fully available. The data underlying the results presented in the study are available from Carlos Brandão (e-mail: Carbrandao@prestadores.amil.com.br) for researchers who meet the criteria for access to confidential data according to the Ethics Committee (contact via Comitê.etica@procardiaco.com.br).

**Funding:** The author(s) received no specific funding for this work.

**Competing interests:** The authors have declared that no competing interests exist.

## Conclusions

NIV was not associated with lower hospital mortality when compared to IMV in critically ill VOP admitted with CAP, but there was no strong signal of harm from its use. The main confounders of this association were both the severity of respiratory dysfunction and of extra-respiratory organ failures.

## Introduction

The very old patients ($\geq$ 80 years-old) (VOPs) are a subpopulation increasingly admitted to ICUs [1]. Common reasons for admission among VOPs are respiratory diagnoses [2], especially pneumonia, for which the initial strategy of respiratory support is controversial: while noninvasive ventilation (NIV) may be an option [3], it carries a risk of failure of up to 50% in this scenario [4, 5], which is associated with worse outcomes [6]. In contrast, invasive mechanical ventilation (IMV) and its downstream consequences such as sedation also carries the risk of important adverse events that could impact both mortality and functional outcomes.

The VOPs commonly have a high burden of comorbidities combined with impaired functional status [2, 7, 8] and frailty [2, 9], which pose a special challenge for the clinician upon treatment decision-making [10]. Indeed, VOP is prone to worse physiological abnormalities and higher risk of NIV failure [11]. By contrast, chronic obstructive pulmonary disease (COPD) and congestive heart failure (CHF) are common comorbidities in VOP and are known to benefit from NIV in acute respiratory failure [3, 12, 13], possibly even in patients with pneumonia [3].

The decision to provide organ support in VOPs involves more complex decision-making and may vary among intensivists [14]. While the elderly are reluctant to accept IMV as a reasonable life sustaining therapy [15], NIV may still be considered a reasonable option as ceiling therapy without detrimental effects on quality of life [16]. In this context, NIV in VOPs may allow for a less invasive strategy in a time-limited ICU trial [17, 18], which could suffice for decision making throughout the course of ICU stay while not lending these patients subject to unnecessary suffering and potential harms from IMV. However, data supporting the use of NIV in this subpopulation is scarce and recent research demonstrates sustained mortality rates despite antimicrobial availability (in pneumococcal pneumonia) and suggests non-antimicrobial strategies should be further evaluated [19].

Our objective was to evaluate the association between the initial ventilatory strategy of respiratory support–NIV or IMV–and hospital mortality in a representative sample of VOPs admitted to intensive care units with respiratory failure from community-acquired pneumonia (CAP) as main reason for ICU admission. We hypothesized that a less invasive strategy of organ support (NIV) would be beneficial in VOPs with pneumonia.

## Material and methods

### Study design, setting and ethical considerations

This is a multicenter retrospective cohort study between January, 2009 and December, 2012. There were 11 participating ICUs from a Brazilian network of private hospitals. One hospital is specialized in the care of heart diseases, while the others are mixed ICUs.

The Research and Ethics Committee of Hospital Pró-Cardíaco–the reference ethics committee designated by the National Research Ethics Committee–approved the retrospective

analysis and publication of the data under the number 729.008 (CAAE: 33111214.3.0000.5533) and waived the need for informed consent given the retrospective nature of this study. We adhered to the STROBE guidelines (S1 File) and to the guidance from editors of respiratory, critical care and sleep journals [20].

## Study population, exposures and outcome

The study population comprised very old patients ($\geq$ 80 years old) admitted to the intensive care unit with CAP as the main reason for admission for whom invasive or noninvasive ventilation was deemed necessary. We excluded ICU readmissions. The exposure of interest was whether the initial respiratory support was NIV or IMV in the first day of ICU admission. The primary outcome was hospital mortality.

## Data collection and definitions

We retrieved data from a prospectively collected multicenter ICU database (Epimed Monitor System®, Epimed Solutions®, Rio de Janeiro, Brazil), a cloud-based registry for ICU quality improvement in Brazil [21]. Data of all admitted patients are entered in the system by a trained case manager nurse and regularly audited. Retrieved variables included demographics (age and sex), body mass index, admission SAPS 3 [22], $1^{st}$ day sequential organ failure assessment (SOFA) score [23], previous functional status [7], comorbidities (all those from the Charlson comorbidity index) [24], $1^{st}$ hour physiological data ($P_aO_2/F_iO_2$ ratio, $P_aCO_2$ and pH), use of organ support in the $1^{st}$ hour, $1^{st}$ 24 hours and throughout ICU stay (vasopressors, renal replacement therapy, IMV, NIV), ICU and hospital length-of-stay and mortality, and palliative care decision within 24 hours of ICU admission. We calculated the Pneumonia severity index (PSI) [25], the modified frailty index (MFI) [26] and whether the patients were septic or not (according to Sepsis 3.0 definitions) [27, 28] from available variables of the dataset.

## Data analysis

Categorical variables are described as numbers and percentages. Quantitative variables are presented as means (standard deviations) or medians [$25^{th}$ percentile, $75^{th}$ percentile] accordingly. Continuous variables were evaluated for normality with histograms and analyzed with t-tests when appropriate. Non-normal continuous variables and discrete variables were analyzed with the Wilcoxon rank-sum test. Categorical variables were analyzed with the chi-squared test or the Fisher exact test, as appropriate.

The choice of the initial treatment strategy is associated with several clinical conditions. As recommended by Lederer et al., we chose the set of potential confounding factors based on a directed acyclic graph (DAG), accounting for causal paths and avoiding mediators, open back-door paths and collider-bias [20] (S1 Fig). We ran multivariable logistic regression models adjusting for potential confounders with different sets of covariates at a time: model 1: age and sex; model 2: model 1 plus body mass index (a proxy for malnutrition), COPD, heart failure, dementia and performance status; model 3: model 2 plus non-respiratory SOFA and admission source; model 4 (main model): model 3 plus respiratory variables ($P_aO_2/F_iO_2 < 150$ mmHg, pH $< 7.3$, $P_aCO_2 > 50$ mmHg).

We explored the robustness of the main model with four sensitivity analyses: 1) adding the SAPS 3 score in model 2, providing a different strategy to deal with severity and acuity variables spending less degrees of freedom; 2) excluding patients in whom palliative care was ascertained in the first day of ICU admission; 3) analyzing only patients admitted directly from the emergency department; and 4) running the main model (model 4) in complete case analysis. We also performed a post-hoc survival analysis censoring follow-up at 28 days of ICU

admission. For this analysis, since we followed-up patients up to hospital discharge (or death), which would lead to informative censoring, we considered patients discharged alive as alive up to 28 days. We present results by their primary group (NIV vs. IMV) with a crude analysis (Kaplan-Meier survival curve and hazard ratios with 95% CIs from a univariate Cox model) and an adjusted analysis in a multivariable Cox model (adjusting for age, sex, modified frailty index and SAPS 3). We assessed proportional hazards by Schoenfeld residuals and log-log plots.

We evaluated effect modification by adding an interaction term in model 4 (main model) for ad-hoc defined subgroups: COPD, heart failure, severe functional status impairment, non-respiratory SOFA $\geq$ 4, $P_ao_2/F_io_2 \leq 150$ mmHg and acute respiratory acidosis (pH < 7.3 and $P_aco_2 \geq 50$ mmHg). Cut-off values were based on standard values used in other studies and traditional indications for IMV (eg, acute respiratory acidosis).

We derived the marginal prediction from model 4 to illustrate the association between the initial strategy of ventilation and mortality in some scenarios. The marginal prediction represents the mortality predicted by the model if all patients in the cohort had received NIV or IMV in each scenario, while all other covariates are kept as observed [29]. We explored scenarios that contrasted the magnitude of extrapulmonary organ dysfunction [low (0), intermediate (4) vs. high (8) non-respiratory SOFA score]; functional status impairment (severe vs. non-severe); and hypoxemia ($P_ao_2/F_io_2$ ratio > 150 or $\leq$ 150 mmHg). Furthermore, in an as-treated analysis, we evaluated the outcomes of patients of the NIV group who were intubated within or after 24 hours of ICU admission, and calculated adjusted odds ratios, adjusting for as model 4.

We conducted multiple imputations to deal with missing data in covariates [30]. We assumed data to be missing at random and imputed 50 datasets using chained equations with predictive mean matching. We included in the imputation model the outcome and exposure variables, all variables considered for covariate adjustment, interaction terms and auxiliary variables [31]. Further details of data missingness and the imputation model are described in the S1 Checklist.

We considered a p-value <0.05 as statistically significant for all analyses and no multiplicity adjustment was done. StataSE® version 16.0 was used for all analyses and the user-written *mimrgns* command was used to generate marginal effects.

## Results

6,318 very old patients were admitted to the ICUs from Jan, 2009, to December, 2012. 678 patients were admitted due to CAP (S2 Fig), out of which 369 presented with respiratory failure (NIV, 232 [63%]; IMV, 137 [37%]) (Fig 1). Overall, median age was 86 [83; 89] years old and there was a higher proportion of female patients (55%). 31% of patients were severely impaired, while 45% were frail according to the MFI. Median SOFA was 5.5 [3; 8] and mean SAPS 3 was 68 (+/- 14). Table 1 describes the main characteristics of the study groups: there were no differences regarding antecedent characteristics; however, the IMV group differed regarding admission source and acuity variables, with higher non-respiratory SOFA scores, higher SAPS 3 scores, lower pH and lower $P_ao_2/F_io_2$ ratio (Table 1). Overall hospital mortality was 55% (204/369): it ranged from 49% (114/232) in patients of the NIV group to 66% (90/137) for patients in the IMV group.

The primary outcome analysis results are presented in Table 2. When comparing NIV and IMV as the initial respiratory support strategy, the sequential adjustment suggested the association was largely confounded by acuity variables. Indeed, the crude odds ratio (OR) was 0.50 (95% CI, 0.33–0.78, p = 0.002), and subsequently 0.70 (95% CI, 0.41–1.20, p = 0.196) for model

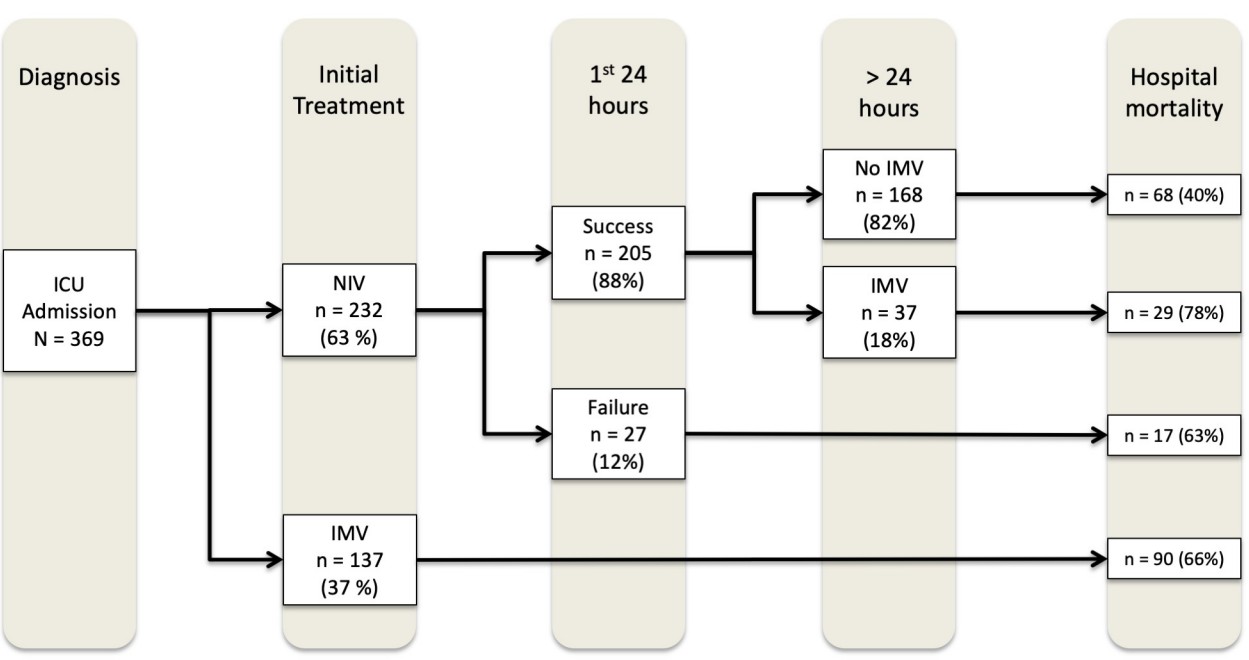

**Fig 1. Flowchart of study participants from intensive care unit admission until the outcome (hospital death or discharge) according to the ventilation strategy.** ICU: Intensive care unit; IMV: Invasive Mechanical ventilation; NIV: Noninvasive ventilation; Failure: need for IMV within 24 hours of ICU admission. Success: no need for IMV within 24 hours of ICU admission.

3 (antecedents plus non-respiratory SOFA) and 0.81 (95% CI, 0.46–1.41, p = 0.452) for model 4 (model 3 plus oxygenation and alveolar ventilation variables).

We observed similar results in sensitivity analyses for the primary outcome comparing NIV vs. IMV (S2 Table): 1) adding SAPS 3 in the model yielded an $_{adj}$OR of 0.77 (95% CI, 0.45–1.31, p = 0.34); 2) excluding patients with an early definition of palliative care (within 24 hours of ICU admission) yielded an $_{adj}$OR of 0.78 (95% CI, 0.44–1.38, p = 0.39); 3) analyzing only patients admitted directly from the emergency department yielded an $_{adj}$OR of 1.03 (95% CI, 0.51–2.08, p = 0.94); 4) analyzing only patients with PSI scores ≥ IV/V yielded an $_{adj}$OR of 0.83 (95% CI, 0.47–1.46, p = 0.51); and 5) the complete case data analysis yielded an $_{adj}$OR of 1.02 (95% CI, 0.40–2.64, p = 0.96). In a survival analysis framework (S3 and S4 Figs), the crude Hazard Ratio (HR) point-estimate was in line with results of Model 1 (HR = 0.75, 95% CI 0.55–1.02, p = 0.065). The adjusted analysis yielded an $_{adj}$HR of 1.19 (95% CI, 0.83–1.69, p = 0.344).

In subgroup analyses (Fig 2), there was no evidence of effect modification for the association between NIV vs. IMV on hospital mortality regarding important antecedent characteristics–performance status, previous diagnosis of CHF and COPD—neither acute physiologic variables–non-respiratory SOFA ≥ 4, acute respiratory acidosis and $P_aO_2/F_iO_2$ ratio < 150 mmHg.

Fig 3 presents the predicted probabilities of hospital mortality in each group across representative values. We observed a small difference in predicted hospital mortality by the initial ventilator strategy. However, there was a sharp increase in predicted mortality for higher non-respiratory SOFA scores. Within each non-respiratory SOFA score group, $P_aO_2/F_iO_2$ ratio ≤ 150 mmHg and, to a lesser extent, severe functional status impaired also changed the marginal probabilities of death.

**Table 1. Patient characteristics at intensive care unit admission.**

| Variable | All patients | NIV * | IMV † | p-value |
|---|---|---|---|---|
| | N = 369 | N = 232 | N = 137 | |
| Age, years | 86 [83; 89] | 87 [83; 90] | 86 [82; 89] | 0.123 |
| Male, n (%) | 167 (45.3%) | 101 (43.5%) | 66 (48.2%) | 0.387 |
| Body mass index < 23 Kg/m$^2$ | 100/252 (39.7%) | 55/154 (35.7%) | 45/98 (45.9%) | 0.107 |
| Charlson comorbidity index | 2 [1; 3] | 2 [1; 3] | 2 [1; 3] | 0.91 |
| Comorbidities | | | | |
| _Hypertension_ | 191 (51.8%) | 123 (53%) | 68 (49.6%) | 0.53 |
| _Diabetes_ | 103 (27.9%) | 60 (25.9%) | 43 (31.4%) | 0.25 |
| _Heart failure_ | 69 (18.7%) | 48 (20.7%) | 21 (15.3%) | 0.202 |
| _COPD_ ‡ | 52 (14.1%) | 28 (12.1%) | 24 (17.5%) | 0.146 |
| _CKD_ $ | 46 (12.5%) | 32 (13.8%) | 14 (10.2%) | 0.32 |
| _Long-term dialysis_ | 7 (1.9%) | 3 (1.3%) | 4 (2.9%) | 0.27 |
| _Cirrhosis_ | 2 (0.5%) | 2 (0.9%) | 0 | 0.53 |
| _Dementia_ | 104 (28.2%) | 65 (28%) | 39 (28.5%) | 0.93 |
| _Cancer_ | 35 (9.5%) | 23 (9.9%) | 12 (8.8%) | 0.71 |
| Performance status impairment | | | | 0.434 |
| _Absent/Minor_ | 118 (32%) | 70 (30.2%) | 48 (35%) | |
| _Moderate_ | 136 (36.9%) | 91 (39.2%) | 45 (32.9%) | |
| _Severe_ | 115 (31.2%) | 71 (30.6%) | 44 (32.1%) | |
| Modified frailty index | | | | 0.985 |
| _Non-frail (MFI $^{\parallel}$ = 0)_ | 27 (7.3%) | 17 (7.3%) | 10 (7.3%) | |
| _Pre-frail (MFI $^{\parallel}$ = 1–2)_ | 176 (47.7%) | 110 (47.4%) | 66 (48.2%) | |
| _Frail (MFI $^{\parallel} \geq$ 3)_ | 166 (45%) | 105 (45.3%) | 61 (44.5%) | |
| PSI ¶ IV/V | 340 (92.1%) | 205 (88.4%) | 135 (98.5%) | < 0.001 |
| Sepsis | 287 (79.7%) | 163 (71.5%) | 124 (93.9%) | < 0.001 |
| SAPS 3 ** | 67.7 (14) | 62.6 (11) | 76.5 (14.2) | < 0.001 |
| SOFA †† | 5.5 [3; 8] | 4 [2; 7] | 8 [6; 11] | < 0.001 |
| _Non-respiratory SOFA_ †† | 4 [2; 7] | 3 [1; 6] | 7 [4; 9] | < 0.001 |
| _Non-respiratory SOFA $\geq$ 4_ †† | 193 (57.1%) | 96/217 (44.2%) | 97/121 (80.2%) | < 0.001 |
| Arterial blood gases | | | | |
| _pH < 7.3_ | 62/241 (25.7%) | 23/129 (17.8%) | 39/112 (34.8%) | 0.003 |
| _$P_aO_2/F_iO_2 \leq 150$_ | 64/208 (30.8%) | 21/102 (20.6%) | 43/106 (40.6%) | 0.002 |
| _Acute respiratory acidosis_ | 39/241 (16.2%) | 15/129 (11.6%) | 24/112 (21.4%) | 0.039 |
| Palliative care within 24h | 25 (6.8%) | 19 (8.2%) | 6 (4.4%) | 0.20 |
| Admission source | | | | 0.015 |
| _Emergency department_ | 225 (61%) | 143 (61.6%) | 82 (59.9%) | |
| _Ward_ | 89 (24.1%) | 64 (27.6%) | 25 (18.3%) | |
| _Other hospital_ | 43 (11.7%) | 19 (8.2%) | 24 (17.5%) | |
| _Other_ | 12 (3.2%) | 6 (2.6%) | 6 (4.4%) | |
| Other organ support | | | | |
| _Vasoactive drugs (24h)_ | 123 (33.3%) | 31 (13.4%) | 92 (67.2%) | < 0.001 |
| _Vasoactive drugs (any)_ | 167 (45.3%) | 62 (26.7%) | 105 (76.6%) | < 0.001 |
| _RRT_ $^i$ (24h) | 5 (1.4%) | 3 (1.3%) | 2 (1.5%) | >0.99 |

(_Continued_)

**Table 1.** (Continued)

| Variable | All patients | NIV [*] | IMV [†] | p-value |
|---|---|---|---|---|
| | N = 369 | N = 232 | N = 137 | |
| RRT [‡] (any) | 22 (6%) | 11 (4.7%) | 11 (8%) | 0.198 |

[*] Noninvasive mechanical ventilation

[†] Invasive mechanical ventilation

[‡] Chronic obstructive pulmonary disease

[§] Chronic kidney disease

[||] Modified frailty index

[¶] Pneumonia severity index

[**] Simplified acute physiologic score 3rd version

[††] Sequential organ failure assessment in the first 24 hours of ICU admission

[‡‡] Renal replacement therapy.

In an as-treated analysis, 27 out of 232 patients (11.6%) of the NIV group were intubated within 24 hours of ICU admission and 37 out of the remaining 205 (18%) were intubated after 24 hours (Fig 1). Patients of the NIV group who did not need IMV during their ICU course had the lower odds of death ($_{adj}$OR 0.52, 95% CI, 0.28–0.97, p = 0.039). By contrast, patients who were intubated after the first 24 hours of ICU admission had higher odds of death ($_{adj}$OR 3.22, 95% CI, 1.21–8.55, p = 0.019) (S3 Table).

## Discussion

### Main findings

In this multicenter cohort study, NIV as the initial ventilatory strategy was not associated with lower hospital mortality among VOPs admitted to the ICU with CAP. We observed a significant degree of positive confounding, mainly explained by acuity variables included in the SOFA score. There was no strong evidence of effect modification regarding important subgroups of patients who usually benefit from NIV (COPD, heart failure) [12, 13] neither on those who usually have worse outcomes under NIV in this setting ($P_aO_2/F_iO_2 < 150$, higher non-respiratory SOFA scores) [4]. Results were robust to sensitivity analyses of the main model and the imputation model assumptions. Irrespective of the initial ventilatory strategy, mortality rates were high for patients who were ultimately intubated, especially in the context

**Table 2. Crude and adjusted odds ratios for hospital mortality between groups[*].**

| | OR (95% CI) | p-value |
|---|---|---|
| *Crude analysis* | 0.50 (0.33–0.78) | 0.002 |
| *Model 1* [†] | 0.47 (0.30–0.74) | 0.001 |
| *Model 2* [‡] | 0.45 (0.29–0.72) | 0.001 |
| *Model 3* [§] | 0.70 (0.41–1.20) | 0.196 |
| *Model 4* [||] | 0.81 (0.46–1.41) | 0.452 |

[*] Invasive mechanical ventilation is the referent group.

[†] Model 1: adjusting for age and sex.

[‡] Model 2: adjusting for model 1 + antecedent characteristics (body mass index + comorbidities + functional status).

[§] Model 3: adjusting for model 2 + non-respiratory SOFA + source.

[||] Model 4: adjusting for model 3 + ABG variables ($P_aO_2/FiO_2$ ratio, pH, $Pco_2$).

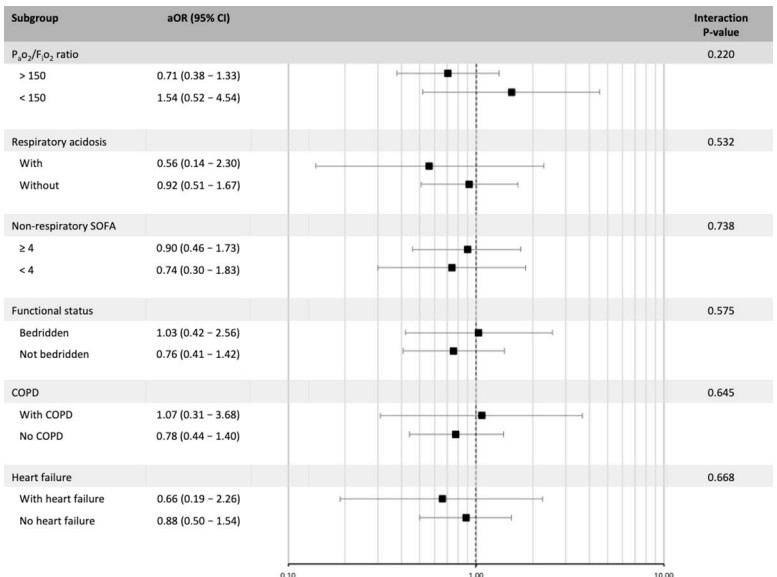

**Fig 2. Subgroup analysis of the effect of noninvasive ventilation vs. invasive mechanical ventilation on hospital mortality.** The x-axis represents the odds ratio for the comparison of noninvasive ventilation (NIV) vs. invasive mechanical ventilation (IMV) according to the analyzed subgroup. SOFA: Sequential organ failure assessment; COPD: Chronic obstructive pulmonary disease.

of high non-respiratory SOFA scores, low $P_aO_2/F_iO_2$ ratio and in patients who were intubated after the first 24 hours of ICU admission. We stress that these results should be interpreted as associations and should not imply causation, since residual confounding may still be an issue.

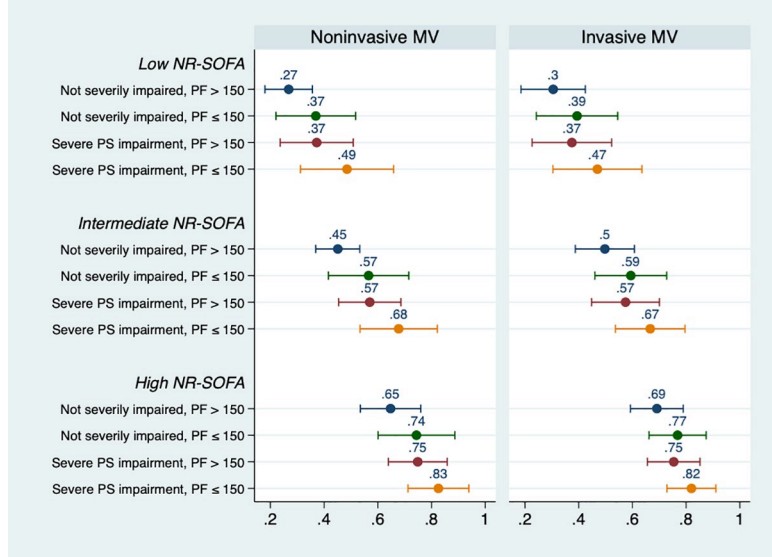

**Fig 3. Marginal probability of hospital mortality according to the initial strategy of mechanical ventilation and three major patient characteristics.** The x-axis represents the probability of the outcome (hospital mortality) according to representative values of covariates, stratified according to the initial strategy of mechanical ventilation (noninvasive ventilation or invasive mechanical ventilation). For example, a patient who was initially under noninvasive ventilation, with an intermediate NR-SOFA, not severely impaired and with a $P_aO_2/F_iO_2$ ratio $\leq$ 150 had a predicted probability of hospital mortality of 57%. MV: mechanical ventilation; PF: $P_aO_2/F_iO_2$ ratio; PS: performance status; NR-SOFA: non-respiratory sequential organ failure assessment; Low-NR SOFA represents a value of 0; intermediate represents a value of 4; high represents a value of 8.

## Relationship with the literature

The hospital mortality for VOPs admitted with CAP to the ICU is quite high. In our cohort, the overall hospital mortality was 55%. In the United States, the 30-day mortality of elderly (≥65 years-old) Medicare beneficiaries admitted with pneumonia to the ICU is of 36% [32], while for those who used NIV or IMV, 30-day mortality was 52.1% and 55.3%, respectively [33]. In the UK case-mix program, data from the early 2000's show a hospital mortality of 50% regardless of age. In the study by Schortgen et al. [34], hospital mortality was 40% and 6-month mortality was 50% for VOPs undergoing NIV irrespective of the initial diagnosis. In this context, we primarily hypothesized that a less invasive strategy would benefit this high mortality target population, but our results were neutral.

One explanation is that the use of NIV in patients with high respiratory drive and high tidal volumes might be a matter of concern [35]. Recent evidence suggests that self-inflicted lung injury may occur through many mechanisms and could be an issue if one decides not to intubate and allow the patient to have substantial spontaneous respiratory efforts [36]. However, noninvasive ventilation is associated with reduced mortality in acute hypoxemic respiratory failure [37] and finding the patient who would be harmed by spontaneous breathing is still a matter of debate, though, including those under IMV [38].

An alternative explanation to our findings is that the strategy of mechanical ventilation may not have an impact on the outcome whatsoever if used within reasonable clinical grounds and considering that NIV is frequently used as ceiling therapy [34]. Valley et al. [33], in a large retrospective cohort of elderly (≥65 years) patients, showed that NIV was no better than IMV, with an absolute risk reduction in 30-day mortality of –0.7% (95% CI, -13.8–12.4, p = 0.92). Our results add to this literature in that we had more granular physiological data beyond ICD-9 codes and we could observe that acuity variables were the main confounders of this association and probably the main determinants of outcome. Although point-estimates did not differ much between NIV and IMV (Fig 3), there was an evident gradient of increased mortality from the confounding variables.

Another concern is that studies assessing NIV in patients with pneumonia have yielded varying results [3, 6, 33, 39], suggesting that NIV could even be harmful if applied to patients with a higher burden of non-respiratory organ dysfunctions or too hypoxemic [4, 5]. Our subgroup analyses assessing effect modification did not suggest this was the case. Point estimates were not worrisome, except for patients with a $P_aO_2/F_iO_2$ ratio ≤ 150, in whom the point estimate differed substantially showing increased harm in those who received NIV, although with a high degree of uncertainty. Although this is biologically plausible and has been shown to be a predictor of failure of NIV, any interpretation of these findings should be very cautious, given the usual low power of subgroup analyses.

Our results should also be interpreted accounting for a risk of misdiagnosis of pneumonia [40]. Common explanations for misdiagnosis include both acute exacerbations of COPD or pulmonary edema from congestive heart failure, which may be difficult to differentiate from pneumonia at patient presentation. Although misdiagnosis could have a potential to bias our results, it would probably do so towards positive associations favoring NIV, which was not the case in our sample. Having said that, we believe this situation actually reflects usual clinical practice and may enhance the generalizability of our findings [41].

## Implications for practice

Our results suggest that a careful balance of benefits and harms of each strategy and the risks of NIV failure–and worse outcomes when in happens–should be weighed against patient's values and preferences to decide the best course of action. Clinicians should not avoid using NIV as an initial strategy of ventilation if it is used within reasonable clinical grounds, since there

was no strong signal of harm from this strategy. Furthermore, NIV use as ceiling therapy should still be regarded as an alternative to IMV when treatment limitations are in place. By contrast, clinicians should not avoid offering IMV for VOPs admitted with pneumonia, especially early during their ICU course. Nevertheless, the prognostic implications of the need for IMV should be considered, especially later during ICU stay and when the burden of extra-pulmonary organ dysfunction is higher and functional impairment is severe.

## Strengths and limitations

Our manuscript has some strengths. The multicenter nature of our study with 11 ICUs from different hospitals and the real-world scenario with treatment limitations in place enhances the generalizability of our results. To account for the inherent risks of an observational study, we developed a causal DAG, which is an important step towards stronger inference and explicit selection of confounders [20], with sensitivity analyses to assumptions of the model being robust. Furthermore, we used multiple imputation to account for data missingness, a recommended approach that usually leads to less biased estimates while retaining higher statistical power [30, 31].

These results are also amenable to limitations. Although we collected data from 11 ICUs over four years, we might have had low power to detect an association of benefit for NIV over IMV, considering we observed a protective point-estimate on our adjusted models; however, we had enough data points to include the main confounders in the regression model without overfitting the data. Our data dates back to 2009–2012, which could be an issue, but except for prone positioning in severe acute respiratory distress syndrome [42], no other therapies appliable to very elderly patients [43] received a strong recommendation since then. We did not have data on pneumonia etiology and specific treatment, which might have an impact on outcome; nevertheless, we do not believe this would explicitly influence clinician's decision to use NIV or IMV and would therefore not be in the DAG. We also did not have access to how NIV or IMV were specifically deployed, such as NIV mode and interfaces and how PEEP was titrated. Furthermore, there could be variability between hospitals because of expertise with NIV, which was not specifically addressed in our models. Hospital mortality may not be the best outcome to be assessed in this population, since long-term functional outcomes could be more valuable; unfortunately, we could not have access to such outcomes. Data on do-not-intubate orders were not explicitly available, although these are seldom used in Brazil and this could be partially captured when excluding patients under palliative care. We also did not have measures of arterial blood gases neither ventilatory variables after the start of NIV, which could provide more mechanistic insights to explain our results [5, 35].

## Conclusions

In a real-world sample of very old patients with community-acquired pneumonia admitted to the intensive care unit, noninvasive ventilation as the initial strategy was not associated with lower hospital mortality when compared to invasive mechanical ventilation, but there was no strong signal of harm from its use. The main confounders of this association were both the severity of respiratory dysfunction and of extrapulmonary organ failures, which should be considered in the decision-making process during the management of very old patients with pneumonia in the intensive care unit.

## Supporting information

**S1 Checklist. STROBE statement checklist.**
(DOCX)

**S1 Fig. Directed acyclic graph.** COPD: Chronic obstructive pulmonary disease; HF: Heart failure; BMI: Body mass index; SOFAnresp: non-respiratory SOFA score; GCS: Glasgow coma

scale. * This directed acyclic graph was built with the online version of DAGitty.
(TIF)

**S2 Fig. Study sample flowchart**\*. * Readmissions excluded from this flowchart.
(TIF)

**S3 Fig. Kaplan-Meier survival plot through 28 days according to initial ventilatory strategy.** Hazard ratio from a Cox proportional hazards model without statistical adjustment.
(TIF)

**S4 Fig. Adjusted survival plot through 28 days according to initial ventilatory strategy.**
Hazard ratio from a Cox proportional hazards model adjusted for age, sex, SAPS3 score and
modified frailty index (mFI).
(TIF)

**S1 Table. Proportion of missing data among variables included in the model.**
(DOCX)

**S2 Table. Sensitivity analyses for the primary outcome.**
(DOCX)

**S3 Table. As-treated analysis for the primary outcome.**
(DOCX)

**S1 File. Additional methods.**
(DOCX)

## Acknowledgments

We would like to thank Leandro Utino Taniguchi, Pedro Caruso and Heraldo Possolo de
Souza for their comments on a previous version of this manuscript. We would also like to
thank Danilo Teixeira Noritomi for his support to start this project.

## Author Contributions

**Conceptualization:** Bruno A. M. P. Besen, Marcelo Park, Otávio T. Ranzani.

**Data curation:** Bruno A. M. P. Besen, Otávio T. Ranzani.

**Formal analysis:** Bruno A. M. P. Besen.

**Investigation:** Bruno A. M. P. Besen.

**Methodology:** Bruno A. M. P. Besen, Otávio T. Ranzani.

**Supervision:** Marcelo Park, Otávio T. Ranzani.

**Visualization:** Bruno A. M. P. Besen.

**Writing – original draft:** Bruno A. M. P. Besen.

**Writing – review & editing:** Bruno A. M. P. Besen, Marcelo Park, Otávio T. Ranzani.

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
