## [Decision Letter · Decision Letter 0]

11 Nov 2020

PONE-D-20-32054

Noninvasive ventilation in very old patients with pneumonia and respiratory failure: a multicenter retrospective cohort study

PLOS ONE

Dear Dr. Besen,

Thank you for submitting your manuscript to PLOS ONE. After careful consideration, we feel that it has merit but does not fully meet PLOS ONE’s publication criteria as it currently stands. Therefore, we invite you to submit a revised version of the manuscript that addresses the points raised during the review process.

 Two authors, experts in the fields, rose several points that I ask you to address in the revised version of the manuscript.

We look forward to receiving your revised manuscript.

Kind regards,

Andrea Cortegiani, M.D.

Academic Editor

PLOS ONE

Journal Requirements:

4. For more information on PLOS ONE's expectations for statistical reporting, please see https://journals.plos.org/plosone/s/submission-guidelines.#loc-statistical-reporting. Please update your Methods and Results sections accordingly.

Reviewers' comments:

Reviewer's Responses to Questions

**Comments to the Author**

1. Is the manuscript technically sound, and do the data support the conclusions?

Reviewer #1: Partly

Reviewer #2: Yes

2. Has the statistical analysis been performed appropriately and rigorously? 

Reviewer #1: N/A

Reviewer #2: Yes

3. Have the authors made all data underlying the findings in their manuscript fully available?

Reviewer #1: Yes

Reviewer #2: Yes

4. Is the manuscript presented in an intelligible fashion and written in standard English?

Reviewer #1: Yes

Reviewer #2: Yes

5. Review Comments to the Author

Reviewer #1: This is a retrospective multicenter cohort study to evaluate the effect of initial mode of ventilation ( NIV vs IMV) in elderly severe CAP with acute respiratory failure. The major limitations of this study are related to data collected retrospectively from a resgistry for ICU quality improvement in Brasil some years ago ( 2009-2012).There is a lack of relevant information that difficult to have solid conclussions: pneumonia etiology, long-term outcomes,limitation of therapy,respiratory mechanics and gas-exchange evolution, standard treament according to new guidelines.Comments:

1. Quality control. Data were collected by trained nurses and regularly audit. Can you please indicate how this was done ? Did you performed a quality control of collected data by a different observer ?

2. Duration of mechanical ventilation. 11.6% of 232 initial NIV patients were intubated within 24 hours and 37/205 (18%) after 24 hours.What was the total duration of IMV and NIV ? A survival analysis during the first 7 days ( Kaplan Meir) adjusted by confounding factors of patients with only IMV, NIV without IMV and NIV with IMV could clarify the impact of NIV alone in outcome.NIV without IMV had an an OR 0.52 and IMV alone an OR 3.22 , both significative.

3. Previous functional status. What was the fragility in this cohort of VOP ? Some authors demonstrated that fragility and initial severity (SOFA) were the most important independent prognostic factors of outcome in VOP admitted to ICU.

4.Severity of acute hypoxic respiratory failure. What was the initial mode of ventilation according to the initial severity of hypoxemia ( mild, moderate and severe determinated by PaO2/FIO2 ) ? What was the short-term evolution of patients with a mild ARF ( PaO2/FIO2 200-300) among both groups with initial NIV or IMV ?

5. What was the tidal volume and driving pressure in boths groups during the first 24-48 hours?Some authors indicated the risk of self-inflicted lung injury during spontaneous ventilation

Reviewer #2: GENERAL COMMENTS

Thank you for allowing me to review this interesting manuscript. This is a multicenter retrospective real-world study on the mortality associated with noninvasive ventilation (NIV) use in elderly patients with community-acquired pneumonia (CAP). The manuscript deals with a poor explored but clinically relevant topic, given the current rise in the number of critically ill elderly patients due to the increased life expectancy and the consequent ethical challenges physicians face regarding aggressiveness of care and resource optimization. Clinicians’ decision about whether to escalate levels of respiratory support is often difficult also due to the modest scientific evidence to guide their choices; therefore, this manuscript may be relevant to clinical practice in supporting clinicians' estimates of patients’ outcomes.

SPECIFIC COMMENTS

The manuscript is well written and easy to read. The quality of written English is acceptable.

Title:

I would add “critically ill” in the title to immediately let the reader understand that the topic is very old patients in ICU settings. I would change as follow: Noninvasive ventilation in critically ill very old patients with pneumonia: a multicenter retrospective cohort study.

Background:

The background is well written and informative. However, I have some suggestions:

1) First sentence needs a reference. I would suggest using the following reference: Laporte L, Hermetet C, Jouan Y, et al. Ten-year trends in intensive care admissions for respiratory infections in the elderly. Ann Intensive Care 2018; 8: 84.

2) I would add a sentence emphasizing the high mortality of patients with CAP (line 56) citing this interesting reference from Plos One: Cillóniz C, Liapikou A, Martin-Loeches I, et al. Twenty-year trend in mortality among hospitalized patients with pneumococcal community-acquired pneumonia. PLoS One 2018; 13: e0200504.

3) Sentence in lines 59-60 needs reference.

4) I would like to point out this interesting and very recent review on the subject that should be considered as a reference in the background: Catia Cillóniz, Cristina Dominedò, Juan M. Pericàs, Diana Rodriguez-Hurtado, Antoni Torres Community-acquired pneumonia in critically ill very old patients: a growing problem. European Respiratory Review Mar 2020, 29 (155) 190126; DOI: 10.1183/16000617.0126-2019

Methods:

The methods used are appropriate for the retrospective design of the study. Study question is clearly stated and clinically relevant. Good data analysis utilizing a proper statistical analysis to answer research questions. However, I have some remarks:

1) There is no mention of the “type of NIV” applied (i.e. modes, interface) and settings, as well as no information on the application and titration of IMV and PEEP. Was low tidal volume ventilation assured to patients treated with IMV? I think that this information, if available, might be relevant since it is proved that adherence to low tidal volume ventilation strategy is important for reducing mortality. If these additional data are not available, this aspect should be acknowledged in the limitation section.

2) Did you record whether or not patients were sedated during NIV? If not, I think it is worth mentioning it, since most patients receiving NIV are managed usually without sedation.

3) Authors collected clinical information on very old patients with CAP treated with NIV or IMV in the past 8 to 11 years. I wonder if the clinical practice on these patients may be very different from recent years in light of the more recent published evidence. Can you comment on that?

4) I think it would be interested to consider differences in time to death between the two ventilator strategies.

Results:

1) My main concern is about the fact that the reference model (model 4) is based on a variable (P/F) that was missing in 161/369 (43.6%), as shown in supplementary table E1.

2) Figure 1: It is not very clear to me. The legend should be more detailed, especially in the definition of NIV Success within 24 h. As it is written now, it looks that definitions of success and failure are the same. Furthermore, in Figure 1: 17 out of 27 patients who fail NIV within the first 24 hours died. What happened to the other 10 patients? Did the patients who fail NIV in the first 24 hours escalate to IMV? This is not clear from the Figure. Also, in Figure 1: Is “No MV” intended as no (invasive or noninvasive) respiratory support? This is not very clear from the Figure and the legend. Please specify. Moreover, in Figure 1 please be consistent in reporting data as N (%) for each field.

Discussion:

The discussion is balanced. References are relevant and updated. Limitations of the study are well addressed by the authors and discussed enough in details, but I would recommend emphasizing some of them better:

1) Since it is a retrospective study, it is impossible to know whether there is a causal relationship between the use of NIV or IMV and mortality. This needs to be more stressed in the discussion.

2) The author should acknowledge that although they attempted to adjust for some case-mix variables, some residual confounding might still be present.

3) Different expertise with NIV among centers should be acknowledged as a possible limitation.

4) I think it is worthy of including and discussing the following references:

- Wood KA, Ely EW. What does it mean to be critically ill and elderly? Curr Opin Crit Care. 2003 Aug;9(4):316-20. doi: 10.1097/00075198-200308000-00011. PMID: 12883288.

- Schortgen F, Follin A, Piccari L, Roche-Campo F, Carteaux G, Taillandier-Heriche E, Krypciak S, Thille AW, Paillaud E, Brochard L. Results of noninvasive ventilation in very old patients. Ann Intensive Care. 2012 Feb 21;2(1):5. doi: 10.1186/2110-5820-2-5. PMID: 22353636; PMCID: PMC3306189.

Minor comments:

Line 55 please add a space before ref 1

Line 78 use reasons for instead of reasons of

Line 93 Please change 80 years-old with 80 years old

Line 94 Please change as main reason with as the main reason

Line 110 pneumonia severity index needs reference, please add the following citation:

Fine MJ, Auble TE, Yealy DM, et al. A prediction rule to identify low-risk patients with community-acquired pneumonia. N Engl J Med 1997; 336(4):243–250.doi:10.1056/NEJM199701233360402

Line 143 change extra pulmonary with extrapulmonary

Line 148 change imputation with imputations

Line 152 change interactions with interaction

Line 156 mimrgns??? Please clarify the typo

Line 161 Please change with years old

Line 255 case mix need an hyphen change into case-mix

Line 290 please add an article before generalizability

Please be consistent throughout the manuscript in referring to invasive mechanical ventilation as IMV.

Please use the term noninvasive ventilation consistently also in the figure legends and avoid non-invasive ventilation

6. PLOS authors have the option to publish the peer review history of their article (what does this mean?). If published, this will include your full peer review and any attached files.

Reviewer #1: No

Reviewer #2: No

---

## [Author Response · Author response to Decision Letter 0]

18 Dec 2020

Reviewer #1: This is a retrospective multicenter cohort study to evaluate the effect of initial mode of ventilation ( NIV vs IMV) in elderly severe CAP with acute respiratory failure. The major limitations of this study are related to data collected retrospectively from a registry for ICU quality improvement in Brazil some years ago (2009 - 2012). There is a lack of relevant information that difficult to have solid conclusions: pneumonia etiology, long-term outcomes, limitation of therapy, respiratory mechanics and gas-exchange evolution, standard treatment according to new guidelines.

Answer: We thank the reviewer for his/her time and effort in reviewing our manuscript and for his/her comments, which helped us improve our manuscript. We agree that the study has limitations, but nevertheless the methodology is as sound as it can be given the limitations of the data and we tempered our conclusions to the results.

Comments:

1. Quality control. Data were collected by trained nurses and regularly audit. Can you please indicate how this was done ? Did you performed a quality control of collected data by a different observer ?

Answer: This is a relevant question. During the deployment of this database in the network of hospitals, each hospital was designated a dedicated case manager nurse to collect data and upload it into the database. This data is therefore collected prospectively. Data was routinely checked for completion and for any relevant discrepancies. Whenever the case manager nurse had any doubts, the ICU director and a centralized chief nurse coordinator were available to clarify any issues. During the training to start data collection, a random check was performed by the ICU director and chief nurse (but we have no data on agreement). We tempered how we framed this in the manuscript, because this is nevertheless a retrospective study and any conclusions should be drawn given this limitation.

2. Duration of mechanical ventilation. 11.6% of 232 initial NIV patients were intubated within 24 hours and 37/205 (18%) after 24 hours. What was the total duration of IMV and NIV? A survival analysis during the first 7 days (Kaplan Meier) adjusted by confounding factors of patients with only IMV, NIV without IMV and NIV with IMV could clarify the impact of NIV alone in outcome. NIV without IMV had an an OR 0.52 and IMV alone an OR 3.22 , both significative.

Answer: Thank you for bringing up this issue. We do not have the total duration of NIV and IMV and unfortunately cannot present this data.

 To account for what both reviewers have asked, we undertook a survival analysis truncated at 28 days. In this analysis, patients discharged alive were considered alive up to 28 days because of the risk of informative right-censoring. We present crude unadjusted Kaplan-Meier survival plots and a survival curve after a Cox model adjusted for age, sex, modified frailty index and SAPS3 score (because there are no missing data in these variables and no imputation model would be necessary). The figures S3 and S4 present the results of these analyses.

3. Previous functional status. What was the fragility in this cohort of VOP ? Some authors demonstrated that fragility and initial severity (SOFA) were the most important independent prognostic factors of outcome in VOP admitted to ICU.

Answer: This is indeed an important consideration. In our dataset, we had available the functional status of our patients as assessed by a three-category performance status categorization. As described in table 1, 31% of this very elderly cohort had severe functional impairment, i.e., they were completely dependent on others for their activities of daily living, while 32% had no functional impairment. This was well balanced between groups, with no statistically significant difference. Given that this is an important confounder, functional status was accounted for in our regression model and was part of our assumptions. Furthermore, although we did not have a frailty measure such as the fragility index or the clinical frailty scale, we derived the modified frailty index (mFI) (table 1), which is also a validated frailty scale to be used specially with database research (1). 45% of our sample was considered frail by the mFI and there was no between group difference in frailty status. Instead of using mFI in our regression model, we included performance status and other covariates (that are part of mFI) in the model to allow exploration of effect modification by subgroup analyses. Regarding the SOFA score, it is an important prognostic factor and it was accounted for in our regression model. To explore both these characteristics, we present in our manuscript the Figure 3, which explores what the reviewer has asked by using marginal probabilities of the outcome (hospital mortality). As the figure shows, P/F ratio < 150, high non-respiratory SOFA scores and performance status impairment all are important characteristics that are associated with increasing hospital mortality regardless of the ventilation strategy.

4. Severity of acute hypoxic respiratory failure. What was the initial mode of ventilation according to the initial severity of hypoxemia (mild, moderate and severe determined by PaO2/FIO2 ) ? What was the short-term evolution of patients with a mild ARF ( PaO2/FIO2 200-300) among both groups with initial NIV or IMV ?

Answer: Thank you for asking for clarification in this issue. We agree that stratifying the analysis according to PaO2/FIO2 ratio is an important issue. However, instead of stratifying in three categories as per ARDS severity (which we cannot state from our data since we cannot ascertain whether infiltrates were bilateral or not), we used the cut-off of 150 in our cohort because this is a common cut-off described for higher risk of NIV failure in the literature, as explained in the manuscript. We refer the reviewer to Figures 2 and 3 of the manuscript. Figure 2 shows a subgroup analysis according to illness severity. Although there is a potential signal of effect modification by P/F ratio (at the cut-off of 150), the results were not statistically significant. Figure 3 presents marginal probabilities of the outcomes given sets of covariates. For example, patients with low non-respiratory SOFA scores and not severely impaired had a probability of death of 27% if their P/F ratio was > 150 and 37% if their P/F ratio was ≤ 150 (in the NIV group). However, for patients in the IMV group, these probabilities were respectively 30% and 39%. Hospital mortality probabilities increase progressively based on performance status impairment, non-respiratory SOFA (a surrogate for increasingly more organ dysfunction) and severity of respiratory dysfunction assessed by the P/F ratio. They are not much different from each initial category of respiratory support. We believe this figure is very informative and can ease the clinical application of our results.

5. What was the tidal volume and driving pressure in both groups during the first 24-48 hours? Some authors indicated the risk of self-inflicted lung injury during spontaneous ventilation

Answer: We agree with the reviewer that these are important issues to be addressed, however these data (tidal volume and driving pressure) are not available in our dataset and neither can be retrieved. This is a limitation of our analysis, which was acknowledged in the limitations session and addressed in the discussion. Self-inflicted lung injury is still a concept under intense discussion in the current literature with no randomized clinical trial data backing up this hypothesis. We also addressed this in the discussion session, while acknowledging that our results still present protective point-estimates of NIV, compatible with a recent meta-analysis published in the literature (2).

 

Reviewer #2: GENERAL COMMENTS

Thank you for allowing me to review this interesting manuscript. This is a multicenter retrospective real-world study on the mortality associated with noninvasive ventilation (NIV) use in elderly patients with community-acquired pneumonia (CAP). The manuscript deals with a poor explored but clinically relevant topic, given the current rise in the number of critically ill elderly patients due to the increased life expectancy and the consequent ethical challenges physicians face regarding aggressiveness of care and resource optimization. Clinicians’ decision about whether to escalate levels of respiratory support is often difficult also due to the modest scientific evidence to guide their choices; therefore, this manuscript may be relevant to clinical practice in supporting clinicians' estimates of patients’ outcomes.

Answer: Thank you for your time and effort in reviewing our manuscript. Your comments allowed us to improve our manuscript substantially. We hope to have addressed them properly and we are open to further clarification if necessary.

SPECIFIC COMMENTS

The manuscript is well written and easy to read. The quality of written English is acceptable.

Answer: Thank you for your comments.

Title:

I would add “critically ill” in the title to immediately let the reader understand that the topic is very old patients in ICU settings. I would change as follow: Noninvasive ventilation in critically ill very old patients with pneumonia: a multicenter retrospective cohort study.

Answer: We agree with the reviewer that this could be a better wording for the title. We changed it accordingly.

Background:

The background is well written and informative. However, I have some suggestions:

1) First sentence needs a reference. I would suggest using the following reference: Laporte L, Hermetet C, Jouan Y, et al. Ten-year trends in intensive care admissions for respiratory infections in the elderly. Ann Intensive Care 2018; 8: 84.

2) I would add a sentence emphasizing the high mortality of patients with CAP (line 56) citing this interesting reference from Plos One: Cillóniz C, Liapikou A, Martin-Loeches I, et al. Twenty-year trend in mortality among hospitalized patients with pneumococcal community-acquired pneumonia. PLoS One 2018; 13: e0200504.

3) Sentence in lines 59-60 needs reference.

4) I would like to point out this interesting and very recent review on the subject that should be considered as a reference in the background: Catia Cillóniz, Cristina Dominedò, Juan M. Pericàs, Diana Rodriguez-Hurtado, Antoni Torres Community-acquired pneumonia in critically ill very old patients: a growing problem. European Respiratory Review Mar 2020, 29 (155) 190126; DOI: 10.1183/16000617.0126-2019

Answer: We have added the references accordingly. They are good uptodate references that further enhance the justification of this study and the knowledge gap to be addressed. Thank you for your suggestions.

Methods:

The methods used are appropriate for the retrospective design of the study. Study question is clearly stated and clinically relevant. Good data analysis utilizing a proper statistical analysis to answer research questions. However, I have some remarks:

Answer: Thank you for your considerations.

1) There is no mention of the “type of NIV” applied (i.e. modes, interface) and settings, as well as no information on the application and titration of IMV and PEEP. Was low tidal volume ventilation assured to patients treated with IMV? I think that this information, if available, might be relevant since it is proved that adherence to low tidal volume ventilation strategy is important for reducing mortality. If these additional data are not available, this aspect should be acknowledged in the limitation section.

Answer: This is indeed an important limitation. We did not have access to such data. We now acknowledged this in the limitations session.

2) Did you record whether or not patients were sedated during NIV? If not, I think it is worth mentioning it, since most patients receiving NIV are managed usually without sedation.

Answer: This is also an interesting issue, but we did not record how and if patients were sedated. However, sedation while on NIV is not a common practice in Brazil and we believe it probably does not influence our results. We therefore have made no changes to the manuscript regarding this comment.

3) Authors collected clinical information on very old patients with CAP treated with NIV or IMV in the past 8 to 11 years. I wonder if the clinical practice on these patients may be very different from recent years in light of the more recent published evidence. Can you comment on that?

Answer: This is indeed an interesting issue and a limitation of our analysis, given the time frame of our database. Although there could be differences in management, only two strategies had confirmatory clinical trial evidence of reduced mortality after 2009-2012, namely prone positioning for moderate-to-severe ARDS and ECMO. ECMO is only rarely deployed, still nowadays, for very elderly patients, given prognostic considerations. Prone positioning, as observed in the ART trial, a Brazilian trial in moderate-to-severe ARDS, was seldom used (10%) within a clinical trial and its uptake was and is still lagging clinical trial evidence. Therefore, we believe that, in spite of being relatively “old” data, our results can still be applied to current practice and to systematic reviews of high-quality observational studies. We added a statement to the limitation session acknowledging this issue.

4) I think it would be interested to consider differences in time to death between the two ventilator strategies.

Answer: Thank you for bringing up this issue. We agree with the reviewer that this is an interesting analysis. Our main concern is that we did not follow-up patients after hospital discharge, which could lead to informative right-censoring in a time-to-event (survival) analysis. Given your suggestion and Reviewer 1 suggestion, we decided to present this data in the manuscript, but in the supplementary material. We truncated the analysis at 28 days and we considered patients discharged alive as alive up to 28 days. We present both a crude analysis (Kaplan Meier survival curve with HR, 95% CIs and p-values from a Cox model) and an adjusted analysis (Survival plot after a multivariable Cox model adjusted for age, sex, modified frailty index and SAPS3 score [variables without any missing data and that fulfilled the proportional hazards assumption]). Relevant changes were done in the methods and results session. They are presented as sensitivity analyses.

Results:

1) My main concern is about the fact that the reference model (model 4) is based on a variable (P/F) that was missing in 161/369 (43.6%), as shown in supplementary table E1.

Answer: We agree with the reviewer that this may be an issue. We decided to present our main analysis as is because it allows further exploration of effect modification and how confounded the association of NIV with outcome is by a subset of relevant variables. We stress that the results of our sensitivity analyses to the multiple imputation analysis (both using (a) SAPS 3 as a global measure of physiological derangement, that includes P/F ratio, creatinine, platelets, Glasgow coma scale and pH; and (b) the complete case analysis) was in line with the primary outcome analyses and therefore does not change the substantive interpretation of our results. Furthermore, data missingness up to 50% is tolerable when using multiple imputation, as recommended by some authors (3, 4). We prefer to maintain the manuscript as is to allow better interpretability of results, but if the reviewer and/or the editor believe that other modelling strategy is preferable, we can change it properly.

2) Figure 1: It is not very clear to me. The legend should be more detailed, especially in the definition of NIV Success within 24 h. As it is written now, it looks that definitions of success and failure are the same. Furthermore, in Figure 1: 17 out of 27 patients who fail NIV within the first 24 hours died. What happened to the other 10 patients? Did the patients who fail NIV in the first 24 hours escalate to IMV? This is not clear from the Figure. Also, in Figure 1: Is “No MV” intended as no (invasive or noninvasive) respiratory support? This is not very clear from the Figure and the legend. Please specify. Moreover, in Figure 1 please be consistent in reporting data as N (%) for each field.

Answer: We have corrected the Figure 1 accordingly, trying to be consistent with the terminology of the manuscript as suggested by other comments. We specify success and failure separately (the definition of failure was the need for IMV in the first 24 hours of ICU stay). We hope the Figure is now clearer to understand. In the case further clarification is necessary, we will be happy to address them properly.

Discussion:

The discussion is balanced. References are relevant and updated. Limitations of the study are well addressed by the authors and discussed enough in details, but I would recommend emphasizing some of them better:

Answer: Thank you for your comments. We believe your considerations are relevant and we have changed the manuscript accordingly to account for them.

1) Since it is a retrospective study, it is impossible to know whether there is a causal relationship between the use of NIV or IMV and mortality. This needs to be more stressed in the discussion.

Answer: We agree with the reviewer. To make a strong message regarding this, the last statement of the statement of main findings now read as follows: “Our results should be interpreted as associations and should not imply causation, since residual confounding may still be an issue.” Furthermore, throughout the manuscript we always describe the results as associations and we refrain from wording that would imply causation. In the “Implications for practice” session, we mainly acknowledge the real-world data results of the strategy and the prognostic implications of confounders and disease course.

2) The author should acknowledge that although they attempted to adjust for some case-mix variables, some residual confounding might still be present.

Answer: We agree with the reviewer and we further acknowledge this issue in the discussion session as described above.

3) Different expertise with NIV among centers should be acknowledged as a possible limitation.

Answer: We agree with the reviewer and we acknowledge this also as a limitation.

4) I think it is worthy of including and discussing the following references:

Answer: We agree with the reviewer that these are good references to be included in the manuscript and enhance its value. We have included them accordingly where they fit better.

Minor comments:

Line 55 please add a space before ref 1

Line 78 use reasons for instead of reasons of

Line 93 Please change 80 years-old with 80 years old

Line 94 Please change as main reason with as the main reason

Line 110 pneumonia severity index needs reference, please add the following citation:

Fine MJ, Auble TE, Yealy DM, et al. A prediction rule to identify low-risk patients with community-acquired pneumonia. N Engl J Med 1997; 336(4):243–250.doi:10.1056/NEJM199701233360402

Line 143 change extra pulmonary with extrapulmonary

Line 148 change imputation with imputations

Line 152 change interactions with interaction

Line 161 Please change with years old

Line 255 case mix need an hyphen change into case-mix

Line 290 please add an article before generalizability

Answer: We have done all changes as suggested by the reviewer. Thank you for looking at our manuscript with such a high scrutiny.

Line 156 mimrgns??? Please clarify the typo

Answer: This is actually not a typo. It’s the Stata package used to derive marginal predictions after multiple imputation (mimrgns). We wrote in the session that it is a user-written command to avoid understanding it as a typo.

Please be consistent throughout the manuscript in referring to invasive mechanical ventilation as IMV.

Please use the term noninvasive ventilation consistently also in the figure legends and avoid non-invasive ventilation

Answer: We have reviewed the manuscript to be consistent in the wording of both expressions. Thank you for reviewing our manuscript with such detail.

 

References

1. Zampieri FG, Iwashyna TJ, Viglianti EM, Taniguchi LU, Viana WN, Costa R, et al. Association of frailty with short-term outcomes, organ support and resource use in critically ill patients. Intensive care medicine. 2018;44(9):1512-20.

2. Ferreyro BL, Angriman F, Munshi L, Del Sorbo L, Ferguson ND, Rochwerg B, et al. Association of Noninvasive Oxygenation Strategies With All-Cause Mortality in Adults With Acute Hypoxemic Respiratory Failure: A Systematic Review and Meta-analysis. JAMA : the journal of the American Medical Association. 2020.

3. Vesin A, Azoulay E, Ruckly S, Vignoud L, Rusinova K, Benoit D, et al. Reporting and handling missing values in clinical studies in intensive care units. Intensive care medicine. 2013;39(8):1396-404.

4. Madley-Dowd P, Hughes R, Tilling K, Heron J. The proportion of missing data should not be used to guide decisions on multiple imputation. J Clin Epidemiol. 2019;110:63-73.

---

## [Decision Letter · Decision Letter 1]

13 Jan 2021

Noninvasive ventilation in critically ill very old patients with pneumonia: a multicenter retrospective cohort study

PONE-D-20-32054R1

Dear Dr. Besen,

We’re pleased to inform you that your manuscript has been judged scientifically suitable for publication and will be formally accepted for publication once it meets all outstanding technical requirements.

Kind regards,

Andrea Cortegiani, M.D.

Academic Editor

PLOS ONE

Additional Editor Comments (optional):

Reviewers' comments:

Reviewer's Responses to Questions

**Comments to the Author**

1. If the authors have adequately addressed your comments raised in a previous round of review and you feel that this manuscript is now acceptable for publication, you may indicate that here to bypass the “Comments to the Author” section, enter your conflict of interest statement in the “Confidential to Editor” section, and submit your "Accept" recommendation.

Reviewer #1: All comments have been addressed

Reviewer #2: All comments have been addressed

2. Is the manuscript technically sound, and do the data support the conclusions?

Reviewer #1: Partly

Reviewer #2: Yes

3. Has the statistical analysis been performed appropriately and rigorously? 

Reviewer #1: Yes

Reviewer #2: Yes

4. Have the authors made all data underlying the findings in their manuscript fully available?

Reviewer #1: Yes

Reviewer #2: Yes

5. Is the manuscript presented in an intelligible fashion and written in standard English?

Reviewer #1: Yes

Reviewer #2: Yes

6. Review Comments to the Author

Reviewer #1: Thank you for answering all comments of reviewers and to improve your manuscript that unfortunately have some lack of information that could be interesting to confirm your conclusions . This was included in the limitations of the manuscript in the discussion section : old vs recent years, Vt, Driving Pressure, how PEEP , mode of ventilation and NIV interface were applied .The study can be the base for a prospective study in the future

Reviewer #2: The authors made the necessary changes and the manuscript has improved. I have no further comments or suggestions.

7. PLOS authors have the option to publish the peer review history of their article (what does this mean?). If published, this will include your full peer review and any attached files.

Reviewer #1: No

Reviewer #2: No

---

## [Editor Report · Acceptance letter]

18 Jan 2021

PONE-D-20-32054R1 

Noninvasive ventilation in critically ill very old patients with pneumonia: a multicenter retrospective cohort study 

Dear Dr. Besen:

I'm pleased to inform you that your manuscript has been deemed suitable for publication in PLOS ONE. Congratulations! Your manuscript is now with our production department. 

Kind regards, 

on behalf of

Dr. Andrea Cortegiani 

Academic Editor

PLOS ONE